# Management practices and their relation to success of Polish SMEs: The empirical verification

**Krzysztof Łobos**[1]*, **Magdalena Wojciech**[2]

**1** WSB University in Wrocław, Wrocław, Poland, **2** Faculty of Mathematics, Department of Mathematical Statistics and Econometrics, Computer Science and Econometrics, University of Zielona Góra, Zielona Góra, Poland

* krzysztof.lobos@wsb.wroclaw.pl

**Citation:** Łobos K, Wojciech M (2021) Management practices and their relation to success of Polish SMEs: The empirical verification. PLoS ONE 16(11): e0259892. https://doi.org/10.1371/journal.pone.0259892

**Data Availability Statement:** The legal owner of the database is Central Statistical Office of Poland and we received access to the database on the basis of individual permission of the President of that governmental agency. We received permission

## Abstract

### Purpose

The aim of the paper is to identify management practices that are characteristic for SMEs that achieve market success measured by their business performance in last their years of their operation analyze the relationships between management practices applied in small and medium-sized enterprises and their success measured by their business performance drawing on the data from 2710 SMEs operating on the Polish market.

### Approach/Methodology/Design

A cluster analysis was used to distinguish homogenous SME groups in view of their management practices. We examined differences between groups in terms of their business performance. The HINoV algorithm allowed six variables to be selected out of 32 management practices chosen initially for testing, with these variables providing the basis for grouping. Modal values and medians were calculated for 17 business performance measures in the three clusters produced. The subsequent analysis of those findings was focused on capturing significant differences.

### Findings

In the group of 2710 Polish SMEs, it was possible to verify that there existed an association between management practices in the field of modern HRM, computer systems supporting management and the company's economic performance, as measured by an increase in net revenue and number of customers over the last three years. In clusters where the above mentioned practices were appreciated, modal and median values of the increase reported in net revenue and number of customers were significantly higher.

### Practical implications

The research has shown that at a time marked by a shortage of highly skilled personnel one should pay particular attention to building an integrated and committed team of workers and

to use the data for research and no permission to pass the data for the third person. It is possible check our permission in Główny Urząd Statystyczny, Warszawa al. Niepodległości 208, 00-925, President: dr Dominik Rozkrut (mail: kancelariaogolnaGUS@stat.gov.pl) The database on the basis of which the statistical analyzes were performed and on the basis of which the results presented in the article were obtained is called „Uwarunkowania rozwoju przedsiębiorczości sektora MŚP". To gain access to this database, a written application must be submitted to the President of the Central Statistical Office, Dr. Dominik Rozkrut. Then it is possible, after considering the application, to gain access to the database without the possibility of identifying specific entities (enterprises). We also obtained such a base, without the possibility of identifying specific enterprises. After obtaining individual access for scientists from the Central Statistical Office, and it is possible after proving that the database will be used only for scientific purposes, it is possible to complete and repeat all the research carried out by the authors of the article. The authors did not receive any special privileges in accessing this data that other researchers would not receive.

**Funding:** The author(s) received no specific funding for this work.

**Competing interests:** The authors have declared that no competing interests exist.

to employee empowerment. The research has also shown that SME managers monitor only a fraction of basic business performance measures, which may prove to be a major risk to SMEs.

## Originality/Value

Previous studies have been largely conducted in a fragmentary manner, i.e. they were concerned with the relationships between the application of some practices (strategic management, BPR, entrepreneurial orientation, monitoring, etc.) and selected business effectiveness measures. In this paper, the research covered SME management practices from a variety of areas which were then compared with the entrepreneurs' assessment as to whether the company's economic condition changed over the last three years. It is also the first attempt in post-socialist economies to identify those SME management practices that are related to better economic results.

## 1. Introduction

The aim of this paper is to identify management practices that are characteristic for those SMEs that achieve market success understood as above-average growth and development in the last three years of their operation drawing on data collected from 2710 SMEs operating on the Polish market. Previous studies have largely been conducted in a fragmentary manner, i.e. they have been concerned with the relationships between the application of one group of practices (strategic management, BPR, entrepreneurial orientation, monitoring, etc.) and selected economic performance measures. In this study, the research covered the SME management practices from a variety of areas comparing them with entrepreneurs' assessment as to whether their company's economic condition changed over the last three years. The considerable number of entrepreneurial determinants included initially in the study subsequently went through a selection process based on the modified HINoV algorithm.

The success of SME enterprises depends on both internal and external factors. They often take the form of the so-called barriers to entrepreneurship development of the following nature: (1) internal (e.g. insufficient skills, resources at enterprise level), (2) administrative/regulatory (e.g. tax systems, complicated laws) and (3) financial laws (mainly access to finance). Research on barriers and determinants of entrepreneurship development has been carried out many times at the level of large economic blocks–for example, the European Union by the European Parliament [1], national, for example for Poland by the PARP state agency–Polish Agency for Enterprise Development [2], and as part of a series of scientific studies for individual SME segments or in a regional perspective, see e.g. [3]. In this article, a positive approach was adopted, as a result of which the focus was only on the internal determinants of the success of SMEs, which take the form of management practices, and not on deficiencies or shortcomings of an internal nature or of barriers or conditions in the environment that are independent of enterprises. Research on management practices as determinants of SMEs success was also carried out, for example, in Great Britain [4]. It shows that SMEs less often apply formal management practices than larger companies, but they facilitate their growth and contribute to increased productivity. HR management practices, such as training and performance-related pay, as well as setting formal performance targets are of the greatest importance. The results of another study indicate that organizations that constantly monitor their processes, set goals in a

systematic manner and control the performance of their employees achieve better results than those that do not apply these practices [5]. In the German economy, a relationship between better results and more intensive use of monitoring practices and incentives for employees was found, but more intensive practice of defining performance targets was not associated with better results [6]. In Poland or other post-communist countries, the authors did not find comprehensive studies on the impact of various SME management practices on their success. These countries are characterized by intense economic catching-up in relation to developed economies with a long free-market tradition. In 2018, Poland joined the group of 25 developed countries according to the FTSE Russell classification. Countries such as Lithuania, Latvia, Estonia, the Czech Republic and Slovenia are also developing dynamically. This results in a research gap–if these countries are somewhat different from the developed free-market economies and at the same time play an increasingly important role in the global economic circulation, it seems reasonable to study management practices and their impact on the success of their SMEs, following the example of developed countries.

The large sample of enterprises participating in the study was possible because of the data gathered throughout the course of a scientific project carried out on a national scale in Poland, whose content manager was one of the authors of this paper (KŁ). An experimental research project titled "*Determinants of the entrepreneurship development in the SME's sector.*" was conducted within the framework of Poland's official statistics by the Centre for Statistics Education run by GUS (Statistics Poland) over the years 2017–2018. The experimental nature of the project was found in its collection of qualitative data, not easily measurable, on enterprise management issues, which had not, up to then, been the practice in Poland's official statistics. Data were collected from SMEs involved in industrial processing, construction industry, trade and services, excluding the following industries: agriculture, forestry, hunting, fishing industry, financial and insurance activities, public administration, national defence and compulsory social security.

## 1.1 Management practices as a source of SME success

For this research to be realized, it was necessary to specify clearly all the groups of management practices applied by the contemporary SMEs which may be associated with their economic success. The study focused solely on internal determinants of success of small and medium-sized enterprises. The selection of these determinants was the result of a review of scientific publications which presented findings produced by the research on entrepreneurial determinants, as well as the focus group studies in which SME entrepreneurs participated and which were conducted under the project *Entrepreneurial development determinants in SME sector*. Nine areas of management practices were distinguished as a result of the above including: entrepreneurial orientation, strategic management, modern HRM (human resource management), modern management methods, market relationships, monitoring practices, computer systems, ITC technologies and networking. Each has an aggregate character, which means that it cannot be measured directly but rather through the variables describing specific management practices.

Entrepreneurial orientation (EO) [7] implies commitment to putting into operation innovative solutions, refreshing and improving the market offer, ability to take risk by implementing as yet unproven solutions (products, services, activities on new markets) and also being more proactive than rivals in terms of exploiting market opportunities. The overall reason why EO is considered so crucial is the ever shortening life-cycle of products themselves as well as that of business models [8].

Strategic management is one of those determinants whose importance for SME success has already been evidenced in earlier empirical research, e.g. [9]. What was primarily investigated

was the impact of the formal strategic management. The findings show clearly that it has a positive impact on performance measures (turnover, costs, profits, EVA, etc.). Additional significance of this research arises from its Central-European economic context. The research also verified the importance of formal strategic planning, as expressed by developing an overall strategy and its components such as, for example, marketing strategy or human resource management.

Modern HRM is a variable used in this study, and is made up mainly of representatives of so called high performance work systems (HPWS). The research [10] shows a clear relationship between HPWS practices and higher work productivity, higher profitability figures and increased ability to generate and implement business innovations. It should be noted that the SMEs were examined in a similar context in that it was in one of the post-socialist countries, namely Romania. The importance of HRM has been further highlighted in numerous other scientific publications, so its position is well-established in the context of SME success, however defined. (see e.g. [4,11]).

Modern management methods, which may comprise such methods as BPR (reengineering), outsourcing, lean management, TQM, ISO systems, can also play some role in SME success. They are among the most popular formal management methods, frequently being taken into consideration in the research concerned with management (see e.g. [12], which is why these methods were also included in this study. It should be stressed, however, that the assumption adopted in this study was that these methods would not be well recognized and applied within the SME group. Furthermore, it is clear that the success enjoyed by those methods is determined by such variables as: high level of innovation, employee empowerment, senior management support with a clear strategic direction it gives to an organization (in terms of strategy and available resources), as illustrated by high-tech companies [13]. For SMEs, fulfilling these conditions may not be that frequent.

The relations with the market mainly involve activities aimed at building cost and quality advantage, as well as seeking to personalize market offer while including customer preferences. In the previous studies some of these variables have been referred to as market orientation [14], a term that was not used in this research. This term denoting market orientation encompassed the well-established practice of responding to customers' needs and organizing activities within the company according to the expectations of the market environment. The variables representing market orientation were linked, in the present study, to the variables referring to competitive advantage factors (cost, quality) and were considered jointly as market relations.

Monitoring practices covering monitoring economic performance, cost, etc. and competitors' activities fall under those management practices which influence development very significantly. Their implementation by enterprises enjoying market success over long-term is independent of the boom-bust cycle. There is considerable empirical evidence to support this thesis, including the evidence from the SME group [4]. In the work just cited, the importance of diverse formal management practices is analysed, including the monitoring practices.

In the study presented further on in the chapter, the variables of the information and computerized character were taken into account as well. They include computer systems and information and communication technology (ICT). The decision to examine them was linked to the peculiar "democratization" of those solutions and their ever greater availability. Even the integrated IT management systems, extremely expensive until recently, are currently becoming increasingly more available also to small and medium-sized enterprises. The speeding up and automation of a multitude of business processes can certainly be attributed to these systems, which is why they ought to be recognized as a determinant of success, not unlike having access to multiple markets and target groups, which has been facilitated by networking technologies.

The study on the importance of information technologies for the competitiveness of SMEs in a strategic context was carried out by [15].

Networking is to be understood as diverse forms of collaboration of enterprises, especially in the field of delivery and distribution. In some areas (e.g. research and development, export activities) these forms can replace the costly competition between enterprises in which case we then deal with the networks comprised of competitive entities. There are reasons to believe that so called cross-organizational networks, which connect suppliers, consumers and other stakeholders, e.g. competitors, lenders and non-business entities, have a positive impact on companies' situation and their market success [16]. Variables describing this type of collaboration were included in the networking factor.

The internal determinants of SME success cited above describe the management system in various cross-sections and to a large extent systemically. They refer to technical, organizational and human aspects. Exploring the internal determinants linked to management which exert an impact on SME success is somewhat a novelty in the standard research on enterprises within the framework of official statistics. In order to build up a picture of the changes certainly unfolding in the enterprises in question, it is necessary to follow up on the research which additionally represents a scientifically valuable source of empirical data covering domestic small and medium-sized enterprises.

Examined further on in the paper, the determinants of success are factors in the same sense as it is understood, for instance, in the factor analysis. In this sense, a factor displays the nature of an aggregate. Perceived like this, it cannot be directly observed and only through directly observable variables that make up the factor can we observe it. For every variable observed directly we deal with a qualitative variable described on the ordinal scale, and hence one cannot talk about their measurement units. Consequently, respondents gave ranks to the variables according to the scope of their application, importance or frequency of occurrence.

## 1.2 SME success

In this study, SME success is understood as an above-average performance in the two main areas, that of the company's growth and development.

Company's growth is to be defined as improvement in all measurable, quantifiable and simultaneously key aspects of its functioning, such as market and distribution, production, basic economic and financial categories, personnel and collaboration with other entities. The examples of growth will thus include increased market share, an increase in the number of markets operated or serviced consumers, increased number of distribution channels, increased production, rise in turnover, an increase in employment and in in the number of the company's cooperating parties and other elements, not listed here. Putting it another way, the company's growth indicates an expansion of the scale and scope of its operations as well as the rise in employment or turnover.

The company's development is to be defined as improving the company's key areas of operations. These areas can only be described through qualitative variables, which are difficult to measure. Development means improvement in such areas as competitive position, economic condition, company's management system, customer service system and relations with customers, market offer, manner of collaboration and cooperation with cooperating parties, and the overall operation efficiency. The effects of development can be exemplified by improved financial situation, improved competitive position, improved market offer, improved management system and innovations in products, processes, organization and marketing.

In the research conducted under the project *Entrepreneurial development determinants in SME sector*, among others by the author of this section of the paper, one of the project

segments centred around the determinants of investments carried out by SMEs. This showed that the biggest impact on investment decisions exerted current evaluation of cost-effectiveness of a particular project. One can therefore suggest that there is a relationship between growth and development, where development is determined by the level of growth. This, in turn, produces a "spiral of interdependencies" leading to ever bigger values of growth and development. As a result bigger and more developed companies grow and develop even faster, which stands in stark contrast to smaller and less developed enterprises. This was revealed by the analysis of structural equations carried out on the same dataset as this article, showing that implicit variable Growth precedes implicit variable Development (https://stat.gov.pl/files/gfx/portalinformacyjny/pl/defaultstronaopisowa/6155/1/1/raport_rozwoj_przedsiebiorczosci_w_sektorze_msp.pdf). One may also encounter a "spiral of interdependencies" which goes downward towards increasingly weaker performance, disinvestments, social and financial problems, ultimately leading to the company's failure. That development brought about by growth has yet another rationale in the SME group. SME entrepreneurs do not have the means and resources that large companies do. SMEs tend to make decisions incrementally (step by step) and not in a synoptic way. In order to be able to see that it makes sense to make an investment, SME companies first need to see sound tangible effects (revenues, customer number), all of which are covered by growth.

## 2. Materials and methods

### 2.1. Research methodology

The primary objective of the research was to discover, among small and medium-sized enterprises, an appropriate structure of classes according to the management practices applied in SMEs. 32 variables were chosen describing cross-sectionally the SME management practices, with these practices falling under the nine groups already outlined above, and which may be considered to be the internal source of success among this class of enterprises. Moreover, other 17 variables were selected which referred to SME success, mainly to SMEs' growth, development as well as to their own assessment of their position against their market competitors.

It was decided that the method for possibly finding co-occurrence of specific management practices applied in SMEs and of their economic success would involve identifying clusters of relatively homogenous enterprises in terms of the management practices and whose level of appreciation for those practices was the same. The next step would be to find out whether there were any differences in the assessment of economic success between those groups. Thus, if we could observe on average a higher level of economic success variables in a group more advanced in terms of the management practices, while spotting weaker results in a group less advanced in the management practices, then the thesis statement suggesting the existence of such a relationship could be considered proven. If no such differences were to be observed, then this thesis pertaining to the Polish SME group would have to be refuted.

The variables to be examined came from the nine areas outlined above and were considered to be success determinants. For greater clarity, they were given the following symbols: entrepreneurial orientation (EO), strategic management (SM), modern human resources management (HRM), modern management methods (MM), market relations (MR), monitoring practices (MP), computer systems (ITS), information and communication technologies (IT) and networking (NET). The list of the variables chosen for the study is presented below (Table 1).

The selection of 32 initially chosen variables was analysed in terms of their discriminatory power for the SME set. Six variables were selected eventually for the cluster analysis, including

**Table 1. SME management practices (as determinants of success).**

| Symbols | Determinants of SMEs success | Symbols | Determinants of SMEs success |
|---|---|---|---|
| EO1 | Actively searching for market opportunities (niches) and rapid decision-making. | MR4 | Conducting periodical survey of consumer satisfaction, evaluating the level of consumers' satisfaction with company's products and services. |
| EO2 | Actively searching for business opportunities on international markets. | MR5 | Seeking to offer lower product prices (goods or services) compared to other competing businesses. |
| EO3 | Seeking to develop and implement innovations. | MR6 | Seeking to provide higher quality products (goods or services) compared to other competing businesses. |
| EO4 | Proclivity for risk-taking, making decisions under conditions of uncertainty. | MR7 | Implementing new business models through e.g. using other more attractive forms of sale, distribution channels or improvements and new value for consumers, which are different and more competitive than market rivals. |
| SM1 | Developing company's strategy for more than three years. | MP1 | Continuous monitoring of market performance and current economic situation through the system of financial indicators and if necessary taking corrective actions. |
| SM2 | Developing quality management strategy. | MP2 | Analysing the competition's moves, which provides basis for drawing conclusions as to the way one's own company operates. |
| SM3 | Developing risk management strategy. | ITS1 | Deploying systems for improving management such as CRM. |
| HRM1 | Favouring teamwork, cooperation, ensuring the building of integrated team. | ITS2 | Deploying systems for improving management such as ERP (enterprise resource planning). |
| HRM2 | Modifying the way the company operates by drawing on employee input on problems encountered, suggestions about customer service, recommendations for improvement (employees can influence decisions about their work). | ITS3 | Deploying systems for improving management such as CMS (content management system). |
| HRM3 | Ensuring continuous updating of employees' knowledge, e.g. by improving employees' skills and education through school or non-school-based forms. | ITS4 | Implementing computer systems supporting human resource management (HRM). |
| HRM4 | Ensuring involvement of entire staff whose work is based on loyalty and trust. | IT1 | Having the company's own website. |
| HRM5 | Continuous flow of information between employees and management (and between work teams). | IT2 | Having the company's own internet store. |
| MM1 | Improving management system through reengineering, outsourcing, Lean management. | IT3 | Running the company's own blog or microblog. |
| MR1 | Building on the dominant role of leadership team (e.g. founders, associates, shareholders) responsible for company's direction of development, implementation of key operations and initiation of new projects. | IT4 | Using social media (e.g. Facebook, Twitter, Instagram). |
| MR2 | Building on and following up family tradition and company image. | IT5 | Using multimedia content sharing websites (e.g. YouTube, Flickr, Picasa, SlideShare). |
| MR3 | Building on personalized relations with consumers. | NET1 | Collaboration in relations of a cooperative nature with other businesses. |

the following: HRM1, HRM2, HRM4, and ITS1, ITS2 and ITS3. In their nature, these determinants of entrepreneurship were like ordinal data expressed on a scale of one to five, as: completely insignificant, rather insignificant, neutral, rather significant, definitely significant (of key importance), where respondents could assess the impact level exerted by the particular factors on the development of their companies and on their success.

On the basis of the answers provided to the six entrepreneurship determinants, a cluster analysis of enterprises was performed while also describing the structure of those groups. 17 success variables were additionally selected for profiling the classes produced out of the three groups marked by the following symbols: company's growth (GE), company's development (DE), competitive position (CPE). These variables were calculated in the section below (Table 2). It was decided that for these variables an ordinal scale was to be used in the questionnaire despite the fact that some of them could be measured. This was prompted by the need to

**Table 2. Dimensions of SMEs success (components of SME's success).**

| Symbols | Components of SME's success | Symbols | Components of SME's success |
|---|---|---|---|
| GE1 | How has the number of employees changed over the last three years of the company's functioning. | CPE1 | What is the assessment of the company's financial resources in relation to its main competitors. |
| GE2 | How has the number of customers serviced changed over the last three years of the company's functioning. | CPE2 | How does the company meet quality standards in relation to its main competitors. |
| GE3 | How has the number of cooperating parties, suppliers changed over the last three years of the company's functioning. | CPE3 | What is the assessment of the company's business experience in relation to its main competitors. |
| GE4 | How has the net revenue value changed over the last three years of the company's functioning. | CPE4 | What is the assessment of the relevant technological equipment and instrumentation in relation to main competitors (including hardware and IT solutions). |
| GE5 | How has the current assets value changed over the last three years of the company's functioning. | CPE5 | What is the assessment of the company's staff (including management) in terms of knowledge, skills, qualifications and experience in relation to main competitors. |
| GE6 | How has the equity value changed over the last three years of the company's functioning. | CPE6 | What is the assessment of innovative solution implementation (e.g. in terms of product, advertising campaign) in relation to main competitors. |
| GE7 | How has the value of investment changed over the last three years of the company's functioning. | CPE7 | How is good work atmosphere, employees' loyalty, employee interpersonal relations ensured in the company in relation to main competitors. |
| DE1 | How has the financial condition changed over the last three years of the company's functioning. | CPE8 | How does partnership cooperation with other enterprises develop in relation to main competitors. |
| DE2 | How has the company's competitive market position changed over the last three years of the company's functioning. | | |

make it easier for respondents to answer the questions. Another reason was that growth in companies of different sizes would not be comparable and one would need to create relative measures. The answers ranging from "significant decline" to "significant increase" referred to the last three years which respondents would find easier to assess in a quantified manner on account of "the freshness of events".

Respondents indicated the level of changes, seen over the last three years arising from questions GE1-GE7 and DE1 and DE2, according to an ordinal scale ranging from significant decline/deterioration to significant increase/considerable improvement. Moreover, the level produced by comparing respondent's own company to its competitors (CPE1 to CPE8 variables) was also specified on an ordinal scale and looked as follows: significantly lower, slightly lower, comparable, slightly higher or considerably higher.

## 2.2. Statistical analysis

The auxiliary purpose of the research was to describe the structure of small and medium-sized enterprises in the field of SME management practices. The assumption was that the group of companies investigated was not homogenous in their perception of entrepreneurship determinants (Table 1) in their companies' operations. As a consequence, companies were grouped using a cluster analysis based on selected variables. Given the ordinal nature of the scale for measuring these traits, the selection of variables for the process of grouping was carried out with the use of the modified HINoV algorithm [17], where adjusted Rand index was applied. In this algorithm variables showing the highest topri values have the strongest contribution to discovering the cluster structure of enterprises. Based on a scree diagram of this index for the initial set of 32 internal determinants of entrepreneurship, six were selected for further analyses.

The SME cluster analysis was performed using the agglomeration method for hierarchical grouping combined with the class means technique for determining the distances between the clusters. The generalized distance measure GDM2 [18] was applied for calculating the

dissimilarity matrix for enterprises within the space of diagnostic variables, namely the characteristics of the management practices measured on an ordinal scale. The number of classes was determined based on Gap and Hartigan indices for classification quality assessment. The results of the enterprise classification into groups were evaluated using the adjusted Rand index, which reflects the stability level of classification. For describing and profiling classes, the following measures of descriptive statistics were applied: mode and median. The distributions of answers to the questions aimed at grouping (entrepreneurship determinants) were presented for each cluster in the form of graphs. In order to explain the differences between the clusters, the level and statistical significance of a monotonic relationship between the paired variables participating in the enterprise grouping were determined for each cluster. Kendall's tau correlation coefficient [19] was used for measuring this correlation. Holm p-value adjustment taking into account the number of tests performed was employed in testing the significance of correlations of the paired variables. The significance testing was performed at 0.05 significance level. Moreover, the profiling of the classes obtained was also performed using the additional 17 variables referring to the enterprise success.

The analyses and graph presentations were conducted using R 4.0.3 program [20] combined mainly with the application of the following packages: clusterSim packages [21], likert [22].

## 3. Results

The research was carried out on the basis of data collected during the implementation of the nationwide project "Conditions for the development of entrepreneurship in the SME sector" in Poland. The obtained information and opinions related to the current situation of the company at the end of 2017 and the beginning of 2018, when the survey was conducted. Additionally, the data for 2016 (e.g. net turnover and the sum of the balance of assets) obtained on the basis of the obligatory ST report (Annual enterprise survey) carried out by the Central Statistical Office in 2017 were used. The subjects of the study were non-financial sector enterprises with 10–249 employees, which operated in 2016 and submitted the SP report "Annual enterprise survey" for 2016 –a total of 72,623 entities. The group of enterprises adopted for the study did not include entities conducting activities classified under sections A (agriculture, forestry, hunting and fishing), K (Financial and insurance activities) and O (Public administration and defense, compulsory social security). In Poland, small enterprises includes enterprises employing from 10 to 49 people, and medium-sized enterprises from 50 to 249 people.

The survey was conducted throughout the country, in the period from 18.12.2017 to 31.01.2018, among 72,600 entrepreneurs. Participation in the survey was voluntary. It was carried out electronically–with the use of a specially developed electronic form called SME and the Reporting Portal of the CSO. In total, 44,900 questionnaires were received from the respondents (including 16,100 questionnaires which were not fully completed), which accounted for 62% of the total number of questionnaires sent. The survey was originally planned as a complete, and not representative, survey. Out of the total number of 44,900 questionnaires the authors selected 2,710 enterprises qualified for the analysis; the large number of rejected records resulted from missing data. This sample was therefore formed as a result of, firstly, a part of the questionnaires not being sent back at all and, secondly, a large number of them being rejected due to the lack of necessary data. The limitation of the research is thus the non-random sampling. According to the authors, this is compensated by a large number of respondents, as the number required for a random representative study would be more than 7 times smaller.

The study encompassed 2710 enterprises of which the majority was made up of small businesses (82%), with medium-sized enterprises accounting for 18%. In the structure of the surveyed enterprises (small and medium-sized) according to the predominant type of activity, the number of entities that conducted service activities was highest, constituting 56.4% (especially in small entities– 59.0%, while in medium entities—45.9%), especially in terms of trade, repair of motor vehicles– 27.7%. There was also a significant share of entities whose declared predominant type of activity was related to construction– 12.2% (small entities– 13.2%, medium entities– 8.0%). Industrial activity was conducted by 31.4% of entities (mainly in the field of industrial processing– 28.1%). In the group of enterprises employing 10–49 people, more entities conducted business in the field of trade and repair of motor vehicles (29.5%) than in the field of industrial processing (25.2%). On the other hand, in the group of enterprises employing 50–249 people the opposite was true–much more entities operated in the field of industrial processing (39.8%) than in the field of wholesale and retail trade and repair of motor vehicles (20.7%).

With a view to getting insight into the structure of the enterprises by having them classified into groups according to their management practices and the profiling of thus produced groups in terms of their economic success, a cluster analysis was performed. To this end, the initial set of 32 ordinal variables went through a selections process using the modified HINoV algorithm. The graph presented below (Fig 1) of topri values led to the selection of those variables, out of the 32 characterizing the management practices (Table 1), which allowed the structure of the population to be identified to the largest extent. The graph allows one to conclude that choosing six out of the 32 initially specified variables describing the management

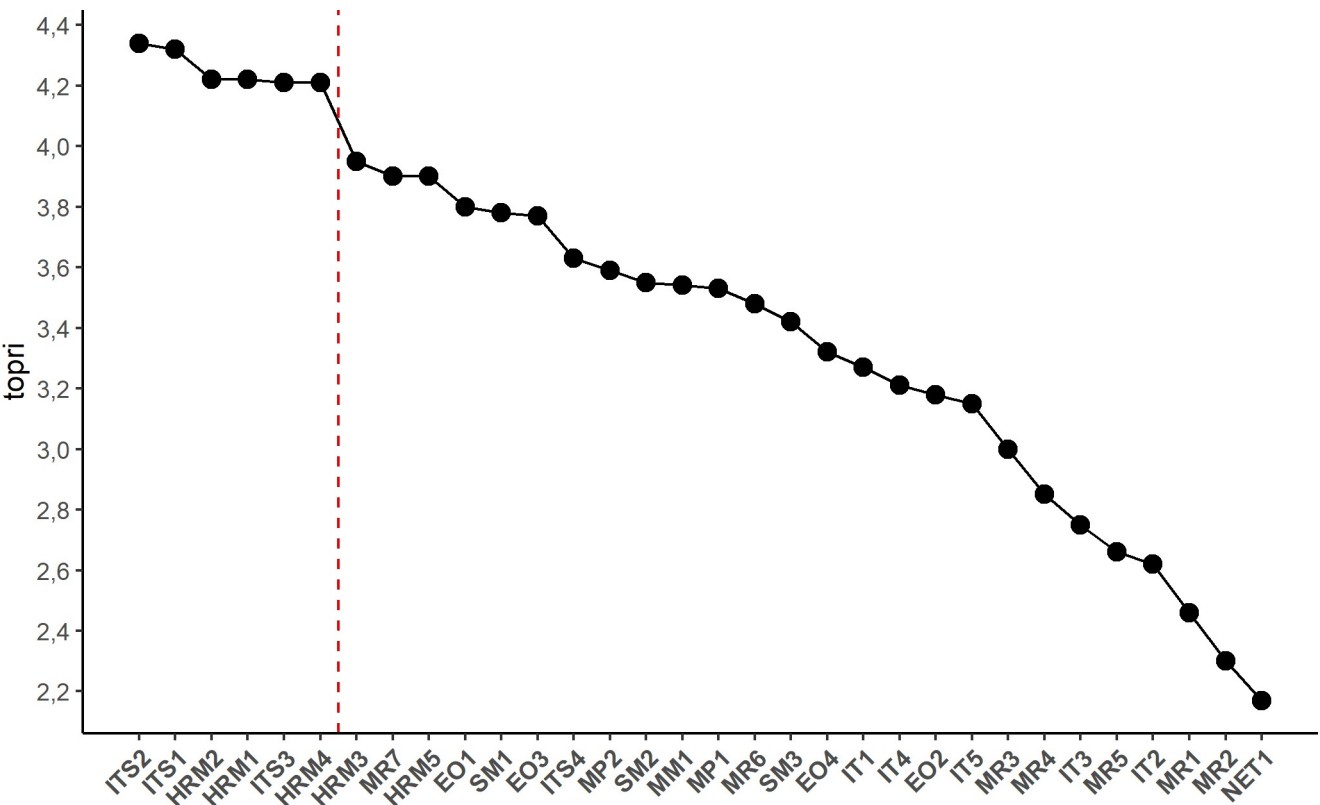

**Fig 1. The graph of the topri index value for the each management practices.**

practices is optimal for further grouping of objects (companies). The variables fall under the group of practices involved in modern human resource management (HRM) and the group of practices associated with the computer systems supporting management (ITS).

The cluster analysis was performed using the agglomeration method for hierarchic grouping combined with the class means technique for determining the distances between the clusters. Considering the ordinal nature of the entrepreneurship determinants, the dissimilarities between the companies were determined using the GDM2 distance measure. The Gap and Harting indices showed that it was appropriate to distinguish three clusters of companies. The results of the enterprise classification into the three groups were evaluated through the adjusted Rand index whose value stood at 74%.

The first cluster covered 1451 companies (54%), the second 1066 (39%) with the last one accounting for the smallest group and consisting of 193 (7%) companies. In the first class, the small enterprises made up 76% of the companies, with medium companies accounting for 24%; in the second and third cluster, these proportions stood at 89% and 11%; and 88% and 12% respectively The distribution of the entrepreneurship determinant values for each cluster is presented in Fig 2. What can be gleaned from this is that in cluster 1 the companies that are in majority are those which assessed the influence of HRM and computer systems (ITS) on the company's development as at least significant. Cluster 2 grouped together the companies which largely recognized the ITS factors as neutral or rather insignificant and completely insignificant, while HRM factors as rather significant and neutral. Moreover, in cluster 3 one can observe the biggest polarization with respect to the significance of HRM and ITS. In this group the HR management practices were considered to be of key importance, while computer systems to be in the main neutral and insignificant.

Additionally, the significance of the monotonic correlation between paired determinants of entrepreneurship was tested for each cluster. Given the ordinal character of the variables, the significance and strength of association between these paired variables was determined based on Kendall's tau correlation coefficient (Fig 3). The correlation analysis allowed the statistically significant correlations to be identified between the categories of the pairs of characteristics participating in the cross-sectional grouping of the three groups.

Cluster 1 encompassed the companies which answered in one accord to all the pairs of the basic questions and all these relationships proved to be statistically significant. Kendall's correlation coefficients show the highest concordance for answers to the following pairs: HRM4 with HRM1 and with HRM2 (the coefficients equalled respectively 0.628 and 0.624); HRM1 with HRM2 (0.613) and ITS1 with ITS2 (0.607). Of further importance is the fact that in this group the associations between the questions from HRM group and ITS were also significant. It can be gleaned from Fig 2 and Table 3 that in this group the highest values of the variable categories were most frequently indicated in the combinations of answers, i.e. a particular factor was considered to be rather significant or of key importance.

The companies from group 2 also perceived the relationships between the factors from the HRM and ITS groups as positive, excluding here the relationship between HRM4 and ITS questions which proved to be statistically insignificant. In this group, a very strong association was found between the paired variables understood as showing high concordance of answers to questions ITS3 with ITS1 and ITS2 (correlations, respectively: 0.778, 0.751), and ITS1 with ITS2 (0.774). From the Table 3 one can infer that in this group the dominant answer combinations were those showing the middle values on the measurement scale, i.e. excluding the two extreme attitudes.

Cluster 3 was made up of the companies which answered the questions both showing concordance (ITS1 i ITS2: 0.740; ITS2 and ITS3: 0.695; ITS1 and ITS3: 0.538; HRM1 and HRM2: 0.253) and discordance (ITS1 i HRM2: -0.213; ITS1 and HRM1: -0.296), with these

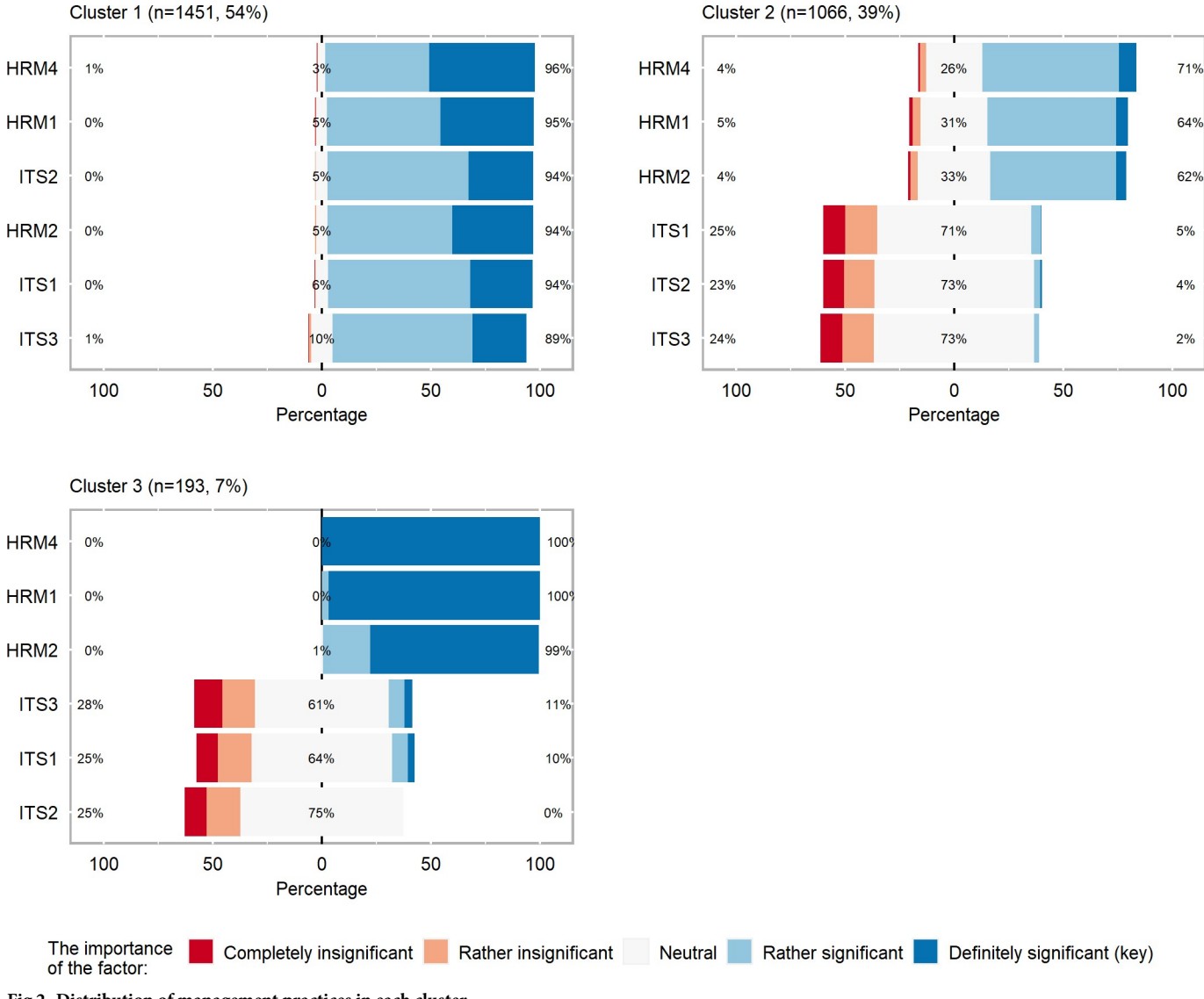

**Fig 2. Distribution of management practices in each cluster.**

associations being statistically significant. This trend of discordant answers can also be noticed in Table 3, where the ITS1 factor was most likely to be indicated as neutral or rather insignificant while simultaneously the HRM1 factor was reported to be of key importance and HRM2 as of key importance or rather significant. One should also emphasize that the role played by the ITS factors in the SME success was most likely to be reported as rather insignificant or neutral. Furthermore, all the companies from the SME group indicated the HRM4 factor as definitely significant- critical for the company's success.

The review of the most frequent combinations of answers regarding the importance of different management practices in SMEs allows one to provide certain synthetic description of these clusters. Clusters 1 and 3 are clusters made up of the companies where the management practices distinguishing companies from one another (this is why they were selected for grouping) are most likely to be appreciated. Moreover, in cluster 1 we could observe that all the management practices, with no exception, were appreciated since respondents assessed them as

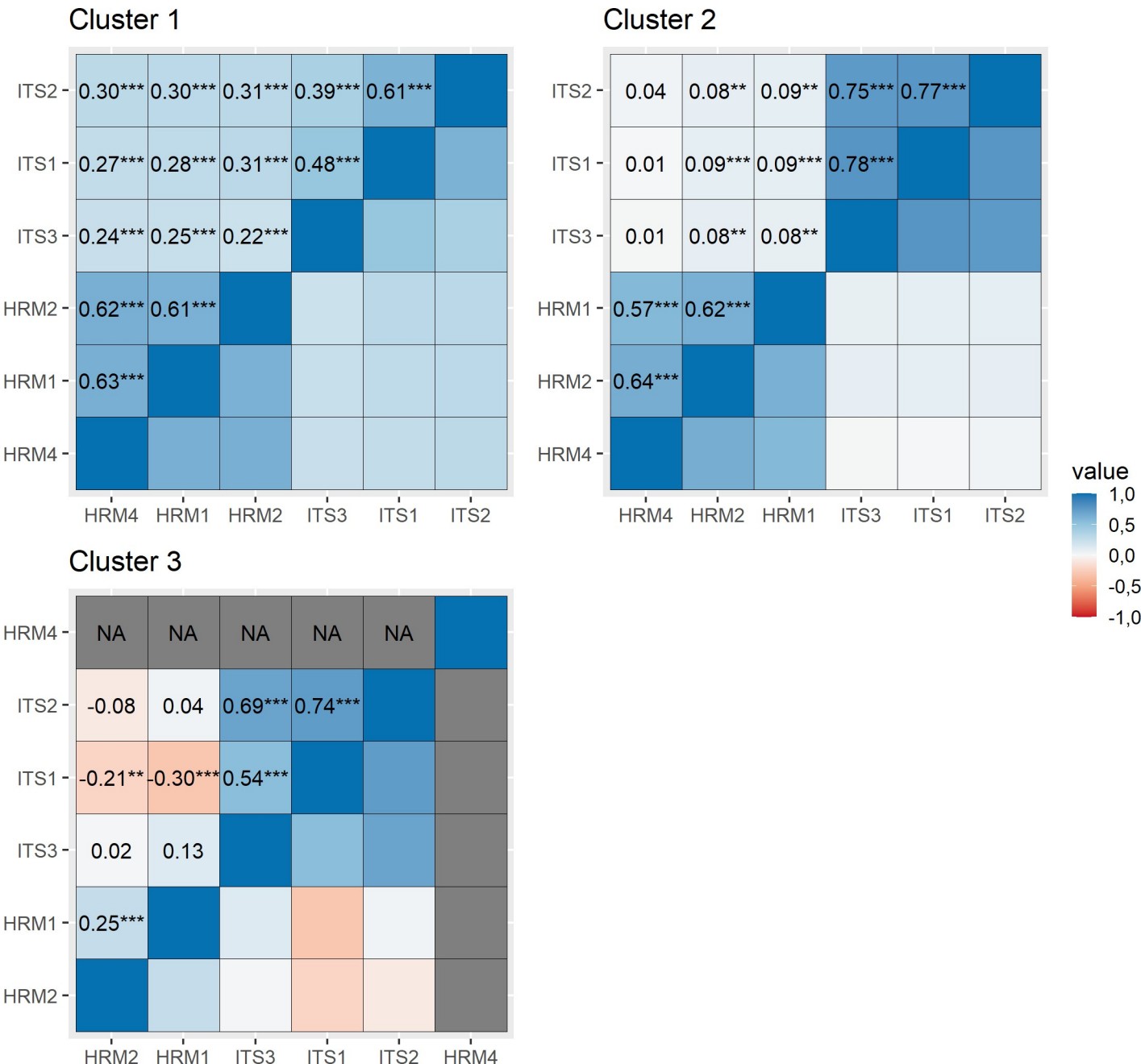

**Fig 3. Kendall rank correlations.** Asterisks indicate statistically significant correlations. In cluster 3, HRM4 has only one value, hence the correlation with this variable is not calculated and has the value NA (not available). Significance codes: ***$p < 0.001$; **$0.001 \leq p < 0.01$; *$0.01 \leq p < 0.05$.

significant or very significant in their business activities (scores 4 or 5), while in cluster 3 we could see this appreciation being selective in that the HRM practices were recognized, while those from the ITS group were not. Cluster 2, on the other hand, displayed a sort of "neutrality" regarding the management practices examined and used in the grouping process by giving them scores close to 3 (with not much significance, insignificant factor). Consequently we could characterize groups 1 and 3 as those recognizing the importance of the modern

**Table 3. The three most frequently indicated by SME response combinations for management practices in each cluster.**

| HRM1 | HRM2 | HRM4 | ITS1 | ITS2 | ITS3 | Number (%) of SME |
|------|------|------|------|------|------|-------------------|
| | | Management practices | | | | Number (%) of SME |
| HRM1 | HRM2 | HRM4 | ITS1 | ITS2 | ITS3 | |
| | | Cluster 1 (n = 1451, 54%) | | | | |
| 4 | 4 | 4 | 4 | 4 | 4 | 368 (25.4%) |
| 5 | 5 | 5 | 5 | 5 | 5 | 164 (11.3%) |
| 5 | 5 | 5 | 4 | 4 | 4 | 116 (8.0%) |
| | | Cluster 2 (n = 1066, 39%) | | | | |
| 4 | 4 | 4 | 3 | 3 | 3 | 301 (28.2%) |
| 3 | 3 | 3 | 3 | 3 | 3 | 184 (17.3%) |
| 4 | 4 | 4 | 2 | 2 | 2 | 46 (4.3%) |
| | | Cluster 3 (n = 193, 7%) | | | | |
| 5 | 5 | 5 | 3 | 3 | 3 | 79 (40.9%) |
| 5 | 4 | 5 | 3 | 3 | 3 | 22 (11.4%) |
| 5 | 5 | 5 | 2 | 2 | 2 | 19 (9.8%) |

management practices, while cluster 2 as one where these practices were found neutral and respondents considered them to be of no practical relevance.

The next step involved comparing the characteristics of the individual clusters with the results produced for the variables referring to the success of the companies (Table 2). Modal values and medians for individual variables describing the components of SME success are presented below (Table 4).

This comparison allows one to conclude that among the 17 dependent variables analysed only for two, namely increased number of customers serviced and an increase in net revenue recorded over the last three years of the company's being in business, can we observe

**Table 4. Modal and medians values for individual variables of SME success.**

| Success measured | Cluster 1 | | Cluster 2 | | Cluster 3 | |
|------------------|-----------|--------|-----------|--------|-----------|--------|
| | Mode | Median | Mode | Median | Mode | Median |
| GE1 | 3 | 3 | 3 | 3 | 3 | 3 |
| GE2 | 3 | 4 | 3 | 3 | 4 | 4 |
| GE3 | 3 | 3 | 3 | 3 | 3 | 3 |
| GE4 | 4 | 4 | 3 | 3 | 4 | 4 |
| GE5 | 3 | 3 | 3 | 3 | 3 | 3 |
| GE6 | 3 | 3 | 3 | 3 | 3 | 3 |
| GE7 | 3 | 3 | 3 | 3 | 3 | 3 |
| DE1 | 3 | 3 | 3 | 3 | 3 | 3 |
| DE2 | 3 | 3 | 3 | 3 | 3 | 3 |
| CPE1 | 3 | 3 | 3 | 3 | 3 | 3 |
| CPE2 | 3 | 3 | 3 | 3 | 3 | 3 |
| CPE3 | 3 | 3 | 3 | 3 | 3 | 3 |
| CPE4 | 3 | 3 | 3 | 3 | 3 | 3 |
| CPE5 | 3 | 3 | 3 | 3 | 3 | 3 |
| CPE6 | 3 | 3 | 3 | 3 | 3 | 3 |
| CPE7 | 3 | 3 | 3 | 3 | 3 | 3 |
| CPE8 | 3 | 3 | 3 | 3 | 3 | 3 |

differences in their modal and median values. For the other variables the values are equal to the middle value (3, nothing changed in this respect). The second observation is as follows: in the clusters characterized by their showing appreciation for the modern management practices the modal values and medians for increased net revenue and an increase in the number of customers stood at 4, which implies their rise. Moreover, in the cluster with the companies seeing those practices as having neutral significance, the analogical values were reported at 3, which means that over the last years those entities saw on average no changes.

## 4. Conclusions

Building on the analysis of the empirical data one can draw the following conclusions:

1. Drawing on the findings one can confirm the statement that in the group of Polish SMEs, entrepreneurs' appreciation for the modern management practices applied in their own companies co-occurs with the company's performance, as measured by an increase in net revenue and number of customers, which both represent a measure of SME success in this study. This may suggest that there is a relationship between modern management practices and SME success in the group of 2710 Polish SMEs included in the study.

2. The SME group is not homogenous when it comes to how the factors ensuring success are perceived. Three groups were distinguished according to the relevance (importance) of the factors defined (included) in the questions from ITS and HRM groups. Two of these groups were larger accounting for 54% and 39%, with one group accounting for 7%. This small group sort of stood out from the other groups in that it showed different trends in their perception of the factors. These were the trends of answers on the ordinal scale both concordant and discordant, unlike in the two other groups, where the answers for every paired variables were always in concordance.

3. The practices which, on the one hand, differentiated the population and on the other were associated with the company's performance were the modern elements of HRM (favouring team work, cooperation, ensuring team integration, modifying the company's operations based on employees' input in terms of problems encountered, suggestions regarding customer service, improvement proposals, ensuring staff involvement whose work is based on loyalty and trust), as well as the application of computer systems supporting management (HRM, CRM, ERP).

4. In the two groups of enterprises described as those which recognize the importance of the modern management practices the modal values and medians of the change observed in net revenue and number of customers over the last three years stood at 4, suggesting an increase, while in the cluster covering the companies which were indifferent towards these practices the analogical values were recorded at 3, in other words, on average no changes occurred over the last three years.

5. Cluster 1 represents a cluster where all the variables illustrating the application of the management practices are at a high level (4,5), while cluster 3 proves to be more selective in this measure, showing 4,5 score only for the practices referring to the modern HRM, while showing no appreciation for the practices from the ITS group.

6. Only two among the 17 dependent variables explored in the study "reacted" to the high scores given to the importance of the modern management practices. One might assume that this is not as much the result of a total lack of impact of the modern management practices on other performance indicators, as it is the fact that Polish SME entrepreneurs

monitor only or predominantly the most basic indicators such as precisely the number of customers or net revenue. This ought to be read as a signal that there is a need for more training and professionalization among this group. Monitoring different performance indicators may, after all, determine corrective actions and hence could impact the company's performance.

## Author Contributions

**Conceptualization:** Krzysztof Łobos.

**Data curation:** Magdalena Wojciech.

**Formal analysis:** Magdalena Wojciech.

**Methodology:** Krzysztof Łobos, Magdalena Wojciech.

**Writing – original draft:** Krzysztof Łobos.

**Writing – review & editing:** Krzysztof Łobos.

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
