## [Decision Letter · Decision Letter 0]

27 Jul 2021

PONE-D-21-15434

Management practices and their relation to success of Polish SMEs. The empirical verification

PLOS ONE

Dear Dr. Łobos,

Thank you for submitting your manuscript to PLOS ONE. After careful consideration, we feel that it has merit but does not fully meet PLOS ONE’s publication criteria as it currently stands. Therefore, we invite you to submit a revised version of the manuscript that addresses the points raised during the review process.

We look forward to receiving your revised manuscript.

Kind regards,

Alessandro Margherita

Academic Editor

PLOS ONE

Journal Requirements:

2. Please note that your manuscript contains several instances of "Error! Reference source not found." For readability, please amend these in your revision.

Reviewers' comments:

Reviewer's Responses to Questions

**Comments to the Author**

1. Is the manuscript technically sound, and do the data support the conclusions?

Reviewer #1: Yes

Reviewer #2: Yes

2. Has the statistical analysis been performed appropriately and rigorously? 

Reviewer #1: Yes

Reviewer #2: Yes

3. Have the authors made all data underlying the findings in their manuscript fully available?

Reviewer #1: Yes

Reviewer #2: No

4. Is the manuscript presented in an intelligible fashion and written in standard English?

Reviewer #1: Yes

Reviewer #2: Yes

5. Review Comments to the Author

Reviewer #1: The article is interesting and relates to the important problem from the point of view of management science. However, despite the quaint approach and a large research sample, it contains some vital shortcomings. They are as follows:

1. The authors do not refer to the research gap existing in the studied area, nor do they point to any literature sources that directly or indirectly refer to the chosen problem. In other words, the authors do not describe the context of the paper and how the study relates to literature published on this topic.

2. The authors do not indicate why and how their study matters.

3. Although the research sample used in this article is extensive, the authors do not indicate how the companies were selected as well as how and when the data were collected.

4. The authors should define SMEs to indicate which entities have been researched in their study. This is important especially when comparing the results to other investigations out of Poland.

5. The authors should refer to the representativeness of the Polish market compared to other countries. Can the conclusions from the Polish market research be extended to other countries?

6. The author should use numbers as in-text citations in the paper.

7. The statement in lines 168-169, “As a result bigger and more developed companies grow and develop even faster, which stands in stark contrast to smaller and less developed enterprises,” is debatable especially in the context of numerous small, dynamically operating start-ups. It should be supported by research results or literature studies.

8. The authors should indicate how they selected variables determining the success of SME operations. In lines 205-206 the authors refer only to items of the literature that are self-citations.

9. The description of the research procedure and the use of 32 variables (e.g., lines 179, 206, 238) is repeated several times in the paper. Please organize the description of the research methods in more transparent and structured way (going from general to details).

10. The section on the research method should be expanded in the article. The clustering process is described very briefly. It would be beneficial to indicate whether other methods of clustering, such as the K-Means clustering, also give the same division results.

11. In the line 290 the authors should insert “medium” instead of “big companies”.

12. Literature studies and references to the existing, up-to-date research on the topic under investigation are very modest. They include mainly the methodological and self-citing items.

13. There are not any limitations of the research indicated.

Reviewer #2: The abstract is partly a repetition of the article. In this sense, it does not meet the conditions set for abstracts.

The author writes several times in the text: "... The aim of this paper is to analyze the relationships between management practices ...". But the question is: Can analysis be an aim or is it a method? There are different points of view, and this should be justified.

The literature review is not complete. It does not contain the latest articles and works. It ends in 2020 (2 articles). A literature review in the context of the aim of the article is its weakness.

Materials and Methods: A lot of technical faults visible in the text as: Error! Reference source not found (11x).

The process of selecting SME companies for the study are required.

The author writes: "... drawing on data collected from 2710 SMEs operating on the Polish market ...". Next: "... the majority was made up of small businesses (82%), with medium-sized enterprises accounting for 18% ...." . In this context the exact structure of the surveyed SMEs is unknown. What is the geographical structure of SMEs? What is the structure of SMEs by industry? On what basis were these SMEs selected for the study? These are key questions - without answers to them, it will be very difficult to relate to the correctness of the conclusions.

As required, each table and drawing must have detailed sources.

The statistical analyzes seems to be correct.

The results are clearly described.

After describing the structure of the surveyed SMEs, the conclusions may be verified.

6. PLOS authors have the option to publish the peer review history of their article (what does this mean?). If published, this will include your full peer review and any attached files.

Reviewer #1: No

Reviewer #2: No

---

## [Author Response · Author response to Decision Letter 0]

11 Oct 2021

Thank you for the insightful reviews to both reviewers. I included all the answers in the rebuttal letter.

Krzysztof Łobos

---

## [Editor Report · Decision Letter 1]

29 Oct 2021

Management practices and their relation to success of Polish SMEs. The empirical verification

PONE-D-21-15434R1

Dear Dr. Łobos,

We’re pleased to inform you that your manuscript has been judged scientifically suitable for publication and will be formally accepted for publication once it meets all outstanding technical requirements.

Kind regards,

Alessandro Margherita

Academic Editor

PLOS ONE
---

## [Editor Report · Acceptance letter]

5 Nov 2021

PONE-D-21-15434R1 

Management practices and their relation to success of Polish SMEs. The empirical verification 

Dear Dr. Łobos:

I'm pleased to inform you that your manuscript has been deemed suitable for publication in PLOS ONE. Congratulations! Your manuscript is now with our production department. 

Kind regards, 

on behalf of

Dr. Alessandro Margherita 

Academic Editor

PLOS ONE